# Can we predict the severe course of COVID-19 - a systematic review and meta-analysis of indicators of clinical outcome?

Stephan Katzenschlager[1☯], Alexandra J. Zimmer[2☯], Claudius Gottschalk[3], Jürgen Grafeneder[4], Stephani Schmitz[3], Sara Kraker[3], Marlene Ganslmeier[3], Amelie Muth[3], Alexander Seitel[5], Lena Maier-Hein[5], Andrea Benedetti[2], Jan Larmann[1], Markus A. Weigand[1], Sean McGrath[6‡], Claudia M. Denkinger[3,7‡]*

1 Department of Anesthesiology, Heidelberg University Hospital, Heidelberg, Germany, 2 Departments of Epidemiology, Biostatistics and Occupational Health, McGill University, Montreal, Canada, 3 Division of Tropical Medicine, Center for Infectious Diseases, Heidelberg University Hospital, Heidelberg, Germany, 4 Department of Clinical Pharmacology, Medical University of Vienna, Vienna, Austria, 5 Division of Computer Assisted Medical Interventions, German Cancer Research Center (DKFZ), Heidelberg, Germany, 6 Department of Biostatistics, Harvard T.H. Chan School of Public Health, Boston, MA, United States of America, 7 German Center for Infection Research (DZIF), partner site Heidelberg, Heidelberg, Germany

☯ These authors contributed equally to this work.
‡ These authors also contributed equally to this work.
* Claudia.Denkinger@uni-heidelberg.de

**Data Availability Statement:** All relevant data are uploaded to Zenodo and publicly accessible via the

## Abstract

### Background

COVID-19 has been reported in over 40million people globally with variable clinical outcomes. In this systematic review and meta-analysis, we assessed demographic, laboratory and clinical indicators as predictors for severe courses of COVID-19.

### Methods

This systematic review was registered at PROSPERO under CRD42020177154. We systematically searched multiple databases (PubMed, Web of Science Core Collection, MedRvix and bioRvix) for publications from December 2019 to May 31st 2020. Random-effects meta-analyses were used to calculate pooled odds ratios and differences of medians between (1) patients admitted to ICU versus non-ICU patients and (2) patients who died versus those who survived. We adapted an existing Cochrane risk-of-bias assessment tool for outcome studies.

### Results

Of 6,702 unique citations, we included 88 articles with 69,762 patients. There was concern for bias across all articles included. Age was strongly associated with mortality with a difference of medians (DoM) of 13.15 years (95% confidence interval (CI) 11.37 to 14.94) between those who died and those who survived. We found a clinically relevant difference between non-survivors and survivors for C-reactive protein (CRP; DoM 69.10 mg/L, CI 50.43 to 87.77), lactate dehydrogenase (LDH; DoM 189.49 U/L, CI 155.00 to 223.98),

following URL: https://zenodo.org/record/5102836#.YPH8vehKg2x.

**Funding:** SM acknowledges support from the National Science Foundation Graduate Research Fellowship Program under Grant No. DGE1745303, National Library Of Medicine of the National Institutes of Health under Award Number T32LM012411, and Fonds de recherche du Québec-Nature et technologies B1X research scholarship. CMD acknowledges the support of the Heidelberg University Hospital. Any opinions, findings, and conclusions or recommendations expressed in this material are those of the author (s) and do not necessarily reflect the views of the funding agencies.

**Competing interests:** I have read the journal's policy and none of the authors of this manuscript have a competing interest.

cardiac troponin I (cTnI; DoM 21.88 pg/mL, CI 9.78 to 33.99) and D-Dimer (DoM 1.29mg/L, CI 0.9 to 1.69). Furthermore, cerebrovascular disease was the co-morbidity most strongly associated with mortality (Odds Ratio 3.45, CI 2.42 to 4.91) and ICU admission (Odds Ratio 5.88, CI 2.35 to 14.73).

## Discussion

This comprehensive meta-analysis found age, cerebrovascular disease, CRP, LDH and cTnI to be the most important risk-factors that predict severe COVID-19 outcomes and will inform clinical scores to support early decision-making.

## Introduction

Coronavirus disease (COVID-19) was declared a pandemic by the World Health Organization (WHO) on March 11[th], 2020 [1]. As of October 31[st] 2020 approximately 46 million people were infected with this virus [2]. The outcomes of COVID-19 vary from completely asymptomatic to hospitalization, ICU admission and death [3, 4].

Several studies aimed to identify possible risk factors for a severe outcome. Studies investigated demographic risk factors and found advanced age to be the strongest predictor of a severe course [5–7]. However, age alone does not explain the variability in the severity of disease with sufficient granularity [8]. Symptoms on presentation associated with severe disease include dyspnea, fever, cough, and fatigue [6, 9, 10]. Several co-morbidities have been identified as risk factors, including cardiovascular disease, obesity, chronic respiratory disease, diabetes, cerebrovascular disease, chronic renal failure and cancer [7, 11–18]. The effect of other co-morbidities on disease outcome remain less clear: e.g. hypertension being associated with a decreased risk [7, 19] for death in some and an increased risk [20] in other publications. Similarly, data on past and current smoking are inconsistent in respect to the association with disease severity [21–25]. Biomarkers predicting severe disease include different markers of inflammation and acute phase reaction (e.g. CRP, procalcitonin (PCT), white blood cells (WBC), lymphopenia, interleukin 6 (IL-6)) [26, 27]. Increased D-dimer levels, as a marker for coagulation and thrombosis, were found to be elevated in non-survivors, whereas other coagulation markers failed to show statistical and clinical difference [13, 28–30]. Markers indicating cardiac damage, such as cardiac troponin I or T and N terminal pro B type natriuretic peptide (NT-proBNP) were also associated with severe disease and mortality [31].

This systematic review aims, to our knowledge for the first time, to comprehensively evaluate demographic, clinical and laboratory indicators for their association with severe COVID-19 and death.

## Methods

This trial was registered at PROSPERO on April 4[th], 2020 (Registration number: CRD42020177154). The PRISMA checklist is provided in the supplementary S2 in S1 File.

### Eligibility criteria

Studies eligible for inclusion provided data on demographic, clinical and/or laboratory risk factors for the following outcomes: hospitalization, intubation, ICU admission, and/or death. Laboratory values and vital parameters taken at hospital admission were considered. Cross-sectional studies, cohort studies, randomized and non-randomized controlled trials were

included. No specific restrictions were placed in terms of demographic and clinical characteristics of the population being studied. The search was conducted on July 29[th], the search date was set from December 1[st] 2019 to May 31[st] 2020.

A full list of data items screened for in the studies is available in the supplementary S12 in S1 File. These data items were chosen, on the one hand, according to the information available in the existing literature and, on the other hand, in order to identify risk factors at hospital admission.

## Search strategy

Medline [PubMed] and Web of Science Core Collection as well as preprint databases (bioRxiv and medRxiv) were searched. The exact search terms were developed with an experienced medical librarian (GG) using combinations of subject headings (when applicable) and text-words for the concepts without language restrictions. The full search strategy used for PubMed is presented in the supplementary S1 in S1 File. The results of the search term were imported into the bibliography manager Zotero (Version 5.0.92) for further processing.

## Study screening and data extraction

Study selection was done by three authors (SK, CG and JG) initially in parallel for five randomly selected papers and after alignment in the selection was guaranteed, it was done independently by each of the reviewers. Article title and abstracts were screened for eligibility in English or German language, followed by a full-text review for those eligible.

A structured electronic data extraction form was developed (AS, LMH, SO, LAS and BP), piloted on five randomly selected papers and then used to extract information from included studies. Six reviewers (SK, CG, SS, SaK, MG and AM) performed data extraction in duplicate for the first five randomly selected papers to ensure alignment and then independently, with concerns being discussed jointly. For continuous indicators we extracted means and standard deviation as well as medians, first quartiles and third quartiles if available. The comprehensive list of data items that were collected is presented in the supplementary S12 in S1 File. Throughout screening and extraction, disagreements were discussed until consensus was reached, and a senior author (CMD) was consulted when necessary.

Given the concern for reporting of the same patients in different publications [32] leading to a bias in the data, we excluded papers which included patients from the same hospital with an overlapping inclusion date. Furthermore, we excluded data from 23 articles (peer-reviewed and preprint), because the reported laboratory values with the reported units were obviously incorrect (supplementary S3 in S1 File), unless we were able to clarify the issue with the authors of the respective paper directly.

## Assessment of study quality

To analyze risk of bias in individual studies, we evaluated the studies using an approach adapted from an existing Cochrane tool by Higgins et al. [33] for systematic reviews that assessed indicators of outcomes. Specifically, we analyzed three areas: 1) case definition and severity definition; 2) patient data availability and exclusions and; 3) selection bias and applicability. We rated the risk of bias in low, intermediate and high risks of bias.

## Statistical analysis

We grouped indicators into binary and continuous indicators across five categories: (1) demographics, (2) symptoms, (3) co-morbidities, (4) laboratory and (5) clinical course/treatment.

We analyzed all available indicators between (1) hospitalized and non-hospitalized patients, (2) ICU-admitted patients and non-ICU admitted patients, (3) intubated and non-intubated patients, and (4) patients who died and patients who survived. Most data were available for ICU admission and death. Thus, we focus on these comparison groups in the main paper and present data on hospitalization and intubation in the supplement. Clinical significance was determined by expert consensus with clinicians. A predefined rule (e.g. 10% above normal range) across biomarkers or vital parameters is not possible as it is different for every marker and the unit chosen.

Meta-analyses were only performed when there were at least 4 primary studies reporting adequate summary data. As the continuous indicators were often skewed and were summarized by medians in most primary studies, we meta-analyzed the difference of medians across groups for continuous indicators. Specifically, we pooled the difference of medians in a random effects meta-analysis using the Quantile Estimation (QE) approach proposed by McGrath et al. [34]. In secondary analyses, median value of indicators in each comparison group were pooled using the same approach.

The QE approach estimates the variance of the (difference of) medians in studies that report the sample median and first and third quartiles of the outcome. When studies report sample means and standard deviations of the outcome, this approach estimates the (difference of) medians and its variance. Then, the standard inverse-variance approach is applied to obtain a pooled estimate of the population (difference of) medians. The population difference of medians can be interpreted as the difference between the median value of the indicator in one group (e.g., those who survived) and the median value of the indicator in the other group (e.g., those who died).

For binary indicators, the pooled odds ratios (OR) and associated 95% confidence intervals (CI) were estimated in a random effects meta-analysis. For both binary and continuous indicators, the restricted maximum likelihood (REML) approach was used to estimate between-study heterogeneity. When REML failed to converge for a continuous indicator, we used the DerSimmonian and Laird (DL) estimator for all analyses involving this indicator.

For all analyses, between-study heterogeneity was assessed by the $I^2$ statistic. The presence of small-study effects was visually assessed in funnel plots. Analyses were performed in R (version 4.0.2) with package 'metamedian' [35] and in Stata (Version 16.1). The code is publicly available on GitHub (https://github.com/stmcg/covid-ma).

## Results

The search resulted in 6,702 articles, of which 3,733 were excluded because they did not present primary data (e.g. guidelines, recommendations, letter to the editors or correspondences, study protocols, modeling), 792 were case reports, 465 focused on patients younger than 18 years and 381 were systematic reviews. In total, 88 articles were included (Fig 1). The majority of studies (52) were conducted in China, 21 in Europe, 12 in the USA, two in Iran, one in South Korea. Most studies were retrospective cohorts (n = 84) and four had a prospective study design. All studies were in English. Data on mortality were reported in 64 studies, data on ICU admission were available in 26 studies (two studies reported both and patients were counted twice). In total, data from 69,762 patients were meta-analyzed, of whom 5,311 died and 57,321 survived and 2,112 provided data on ICU admission while 5,018 did not require ICU admission. We were not able to perform a meta-analysis for all indicators (supplementary S12 in S1 File) extracted from the publications. Meta-analysis for all eligible indicators for each outcome is listed in the supplementary section (S5–S8 in S1 File).

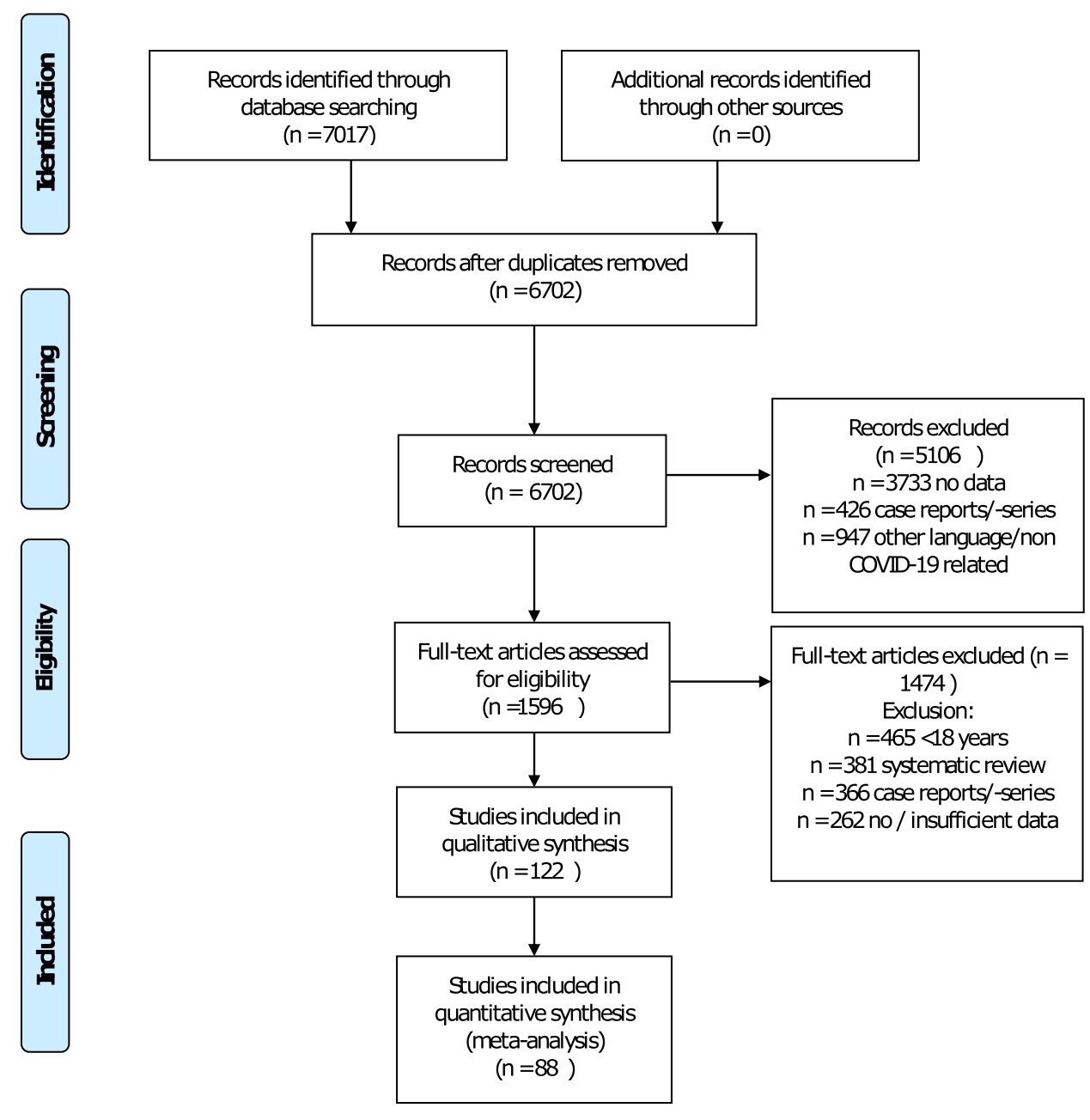

**Fig 1. PRISMA flow diagram.**

## Study quality

The findings on study quality can be found in Fig 2. When considering the case and severity definition of COVID-19, almost 50% of studies were considered low risk of bias, while only 9.1% had a high risk. In contrast, many studies were identified to have high concerns for bias in respect to patient selection and generalizability of findings (36.4% high risk, 9.1% low risk). In more than a third of studies, we had high concern that the full data on patients were not

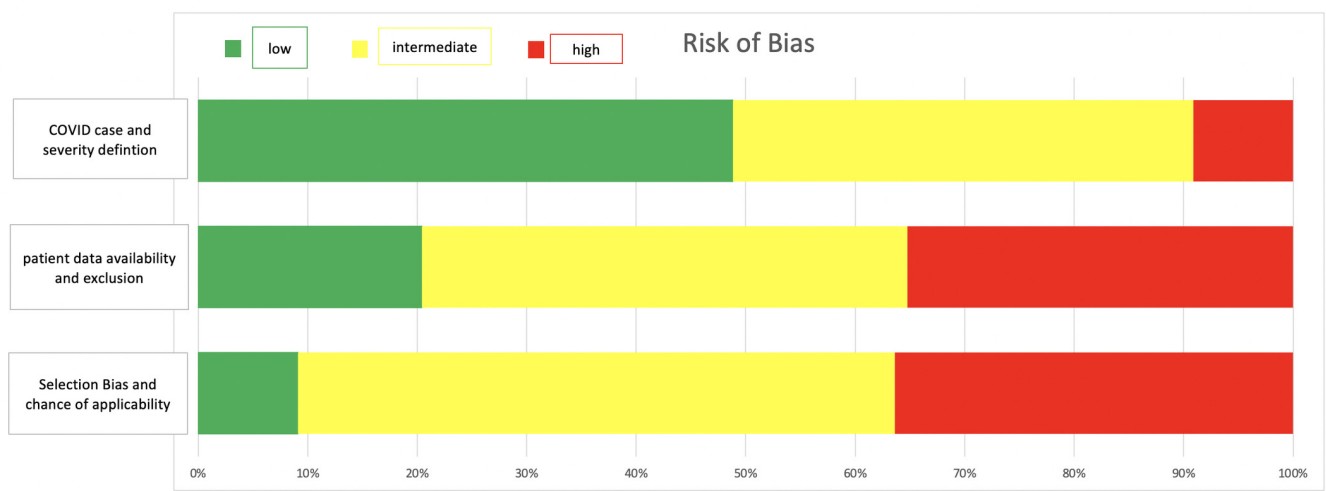

**Fig 2. Risk of bias assessment.**

available and inappropriate exclusion might have occurred (35.2%). The full explanation of the risk of bias assessment and the assessment of each paper individually is available in the supplement S4 in S1 File. Overall, high- or intermediate risk of bias for at least one category was found in almost three fourth (73.8%) of studies. No study scored low risk in all three categories.

## ICU admission

Fig 3 and Table 1 show the pooled odds ratios (OR) and differences of medians (DoM), respectively, for ICU admission for the different indicators in the five categories: demographic, symptoms, comorbidities, laboratory and clinical values [12, 13, 29, 36–57].

Patients requiring ICU admission had a median age of 65 years (CI 62.27 to 66.16). Those not requiring ICU admission were significantly younger with a median age of 59 years (CI 55.93 to 61.86) with a DoM of 4.63 years (CI 1.43 to 7.82) (Table 1). We were not able to perform a subgroup analysis of different age groups as data provided by primary studies was insufficient.

Of the many possible symptoms of COVID-19, we found dyspnea (OR 5.34, CI 2.77 to 10.28) and fatigue (OR 1.63, CI 1.20 to 2.22) to be significantly associated with ICU admission. In terms of co-morbidities, patients admitted to the ICU were more likely to suffer from cerebrovascular disease (OR 5.88, CI 2.35 to 14.73), hypertension (OR 1.62 CI 1.24 to 2.12), diabetes (OR 1.58, CI 1.29 to 1.93) and chronic kidney disease (OR 1.48, CI 1.08 to 2.03). In contrast, cardiovascular diseases (OR 1.50, CI 0.99 to 2.28), chronic obstructive pulmonary disease (COPD) (OR 1.39, CI 0.90 to 2.16), chronic lung disease (OR 1.06, CI 0.89 to 1.25) and smoking (OR 1.00, CI 0.77 to 1.29) were not associated with ICU admission.

Few laboratory values showed differences between patients that required ICU admission and those who did not (Table 1). D-dimer failed to show a statistically significant difference (DoM 0.3 mg/L, CI -0.2 to 0.81). We found a clinically relevant elevation of CRP and cardiac Troponin I (cTnI) in patients requiring ICU admission, although cTnI failed to be statistically significant (DoM for CRP 56.41 mg/L, CI 39.8 to 73.02 and DoM for cTnI 19.27 pg/mL, CI -4.13 to 42.68). A clinically significant reduction in lymphocytes was also observed (DoM -0.34, CI -0.39 to -0.29). Leukocytes, neutrophiles and LDH were also significantly higher in

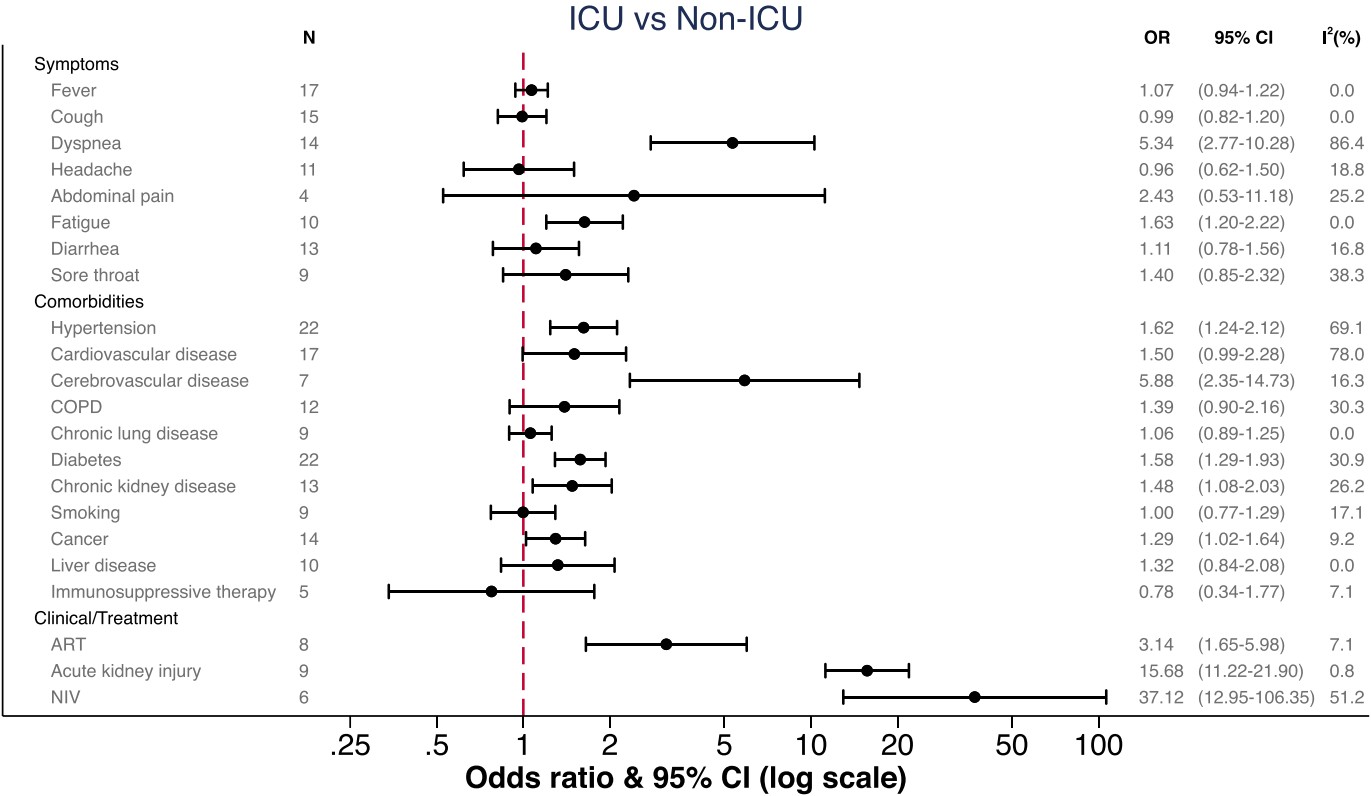

**Fig 3. Pooled odds ratios among ICU vs. non ICU groups.** ICU = Intensive care unit, OR = odds ratio, CI = confidence interval, COPD = chronic obstructive pulmonary disease, ART = anti-retroviral treatment, NIV = non-invasive ventilation.

patients admitted to an ICU, but the absolute elevation over those in non-ICU patients were small and of questionable clinical relevance (Table 1).

Patients developing acute kidney failure, as a complication at any stage, had the highest risk for ICU admission (OR 15.69, CI 11.22 to 21.90).

## Mortality

Fig 4 and Table 2 show the pooled odds ratios and differences of medians, respectively, for mortality for symptoms, comorbidities, laboratory and clinical values [11, 16, 28, 58–110].

Patients who died had a median age of 71 years (CI 69.3 to 71.61) compared to survivors with a median age of 58 years (CI 55.03 to 59.4) for a DoM of 13.15 years (CI 11.37 to 14.94] (Table 2). Again, dyspnea was the symptom that differentiated markedly between survivors and non-survivors (OR 3.69, CI 2.54 to 5.36). Also, fatigue was more frequently observed in those who died (OR 1.48, CI 1.15 to 1.89). Regarding vital parameters at admission, patients who died presented with a median peripheral oxygen saturation (SpO2) on room air of 89% (CI 87.32 to 90.91) to the hospital, while those who survived had 95% (CI 94.59 to 96.63) (DoM -6.33%, CI -8.14 to -4.52).

Patients who died were more likely to suffer from cardiovascular disease (OR 3.93, CI 2.91 to 5.30), cerebrovascular disease (OR 3.45, CI 2.42 to 4.91), chronic lung disease (OR 3.12, CI 2.17 to 4.49), COPD (OR 2.54, CI 1.87 to 3.44; Fig 4) and hypertension (OR 2.49, CI 2.11 to 2.94). Current and former smokers had an increased risk of mortality (OR 1.36, CI 1.10 to 1.67). Patients with chronic kidney disease (CKD) (OR 2.36, CI 1.89 to 2.94), diabetes (OR

**Table 1. Summary of the meta-analysis results for continuous indicators comparing those who were admitted to the ICU and those who were not.**

| Indicator | N. Studies | Pooled DoM [95% CI] | I² |
|---|---|---|---|
| **Demographics** | | | |
| Age (years) | 22 | 4.63 [1.43, 7.82] | 89.89 |
| **Clinical Values** | | | |
| Respiratory Rate (per min) | 5 | 3.15 [0.11, 6.19] | 79.27 |
| **Laboratory Values** | | | |
| Hemoglobin (g/L) | 7 | -5.97 [-11.78, -0.16] | 56.12 |
| Leukocyte (10⁹/L) | 15 | 1.2 [0.54, 1.85] | 62.23 |
| Lymphocyte (10⁹/L) | 19 | -0.26 [-0.34, -0.17] | 75.34 |
| Neutrophil (10⁹/L) | 14 | 2.67 [1.43, 3.91] | 89.14 |
| Platelets (10⁹/L) | 17 | -10.4 [-20.83, 0.04] | 32.66 |
| APTT (sec) | 7 | 0.38 [-1.2, 1.95] | 49.45 |
| D-dimer* (mg/L) | 14 | 0.30 [-0.20, 0.81] | 83.97 |
| Prothrombin (sec) | 7 | 0.48 [0.2, 0.76] | 0.00 |
| ALAT (U/L) | 15 | 4.37 [2.11, 6.64] | 16.17 |
| Albumin (g/L) | 5 | -6.05 [-8.75, -3.35] | 79.38 |
| ASAT (U/L) | 13 | 11.77 [7.24, 16.3] | 64.91 |
| LDH (U/L) | 12 | 140.4 [81.04, 199.76] | 86.32 |
| BUN (mmol/L) | 7 | 1.9 [1.34, 2.45] | 0.00 |
| Creatinine (μmol/L) | 16 | 9.41 [5.18, 13.63] | 40.23 |
| CRP* (mg/L) | 10 | 56.41 [39.8, 73.02] | 76.56 |
| PCT (ng/mL) | 6 | 0.08 [-0.01, 0.16] | 88.76 |
| CK (U/L) | 9 | 33.57 [1.76, 65.38] | 55.08 |
| CK-MB (U/L) | 4 | 2.47 [0.67, 4.26] | 0.00 |
| cTnI* (pg/mL) | 6 | 19.27 [-4.13, 42.68] | 96.82 |

*Indicates that the DL approach was used to estimate between-study heterogeneity. APTT = activated partial thrombin time; ALAT = Alanine transaminase; ASAT = Aspartate transaminase; LDH = Lactate dehydrogenase; BUN = Blood urea nitrogen; CRP = C-reactive protein; PCT = Procalcitonin CK = Creatine kinase; CK-MB = Creatine kinase–myocardial band; TnI = cardiac Troponin I.

2.14, CI 1.82 to 2.52) and cancer (OR 2.08, CI 1.55 to 2.77) also had an increased odds of mortality. Co-morbidities not associated with increased odds of mortality were asthma, liver disease, digestive system disease and immunosuppressive therapy (Fig 4). Clinically relevant elevations outside the normal laboratory range in patients who died compared to those who survived were observed in two markers of inflammation: CRP was elevated by 69.1mg/L (CI 50.43 to 87.77) and IL-6 by 31.19 pg/mL (CI 11.96 to 50.41). Furthermore, clinically significant elevations were observed in cTnI by 21.88pg/mL (CI 9.78 to 33.99) and D-dimer by 1.29mg/L (CI 0.9 to 1.69), while lymphocytes were significantly lower: -0.34x10⁹/L (CI -0.39 to -0.29). Other makers (hemoglobin, leukocytes, neutrophils, platelets, international normalized ratio (INR), Prothrombin, alanine transaminase (ALAT), aspartate transaminase (ASAT), Albumin, LDH, blood urea nitrogen (BUN), Creatinine, PCT, BNP, CK and creatine kinase myocardial band (CK-MB)) were also significantly elevated in those who died. However, the absolute difference compared to those who survived was small and thus likely not clinically relevant. For leukocytes, neutrophils, platelets, prothrombin, ALAT, ASAT, BUN, Creatinine, CK and CK-MB the point estimates even stayed within the normal laboratory range.

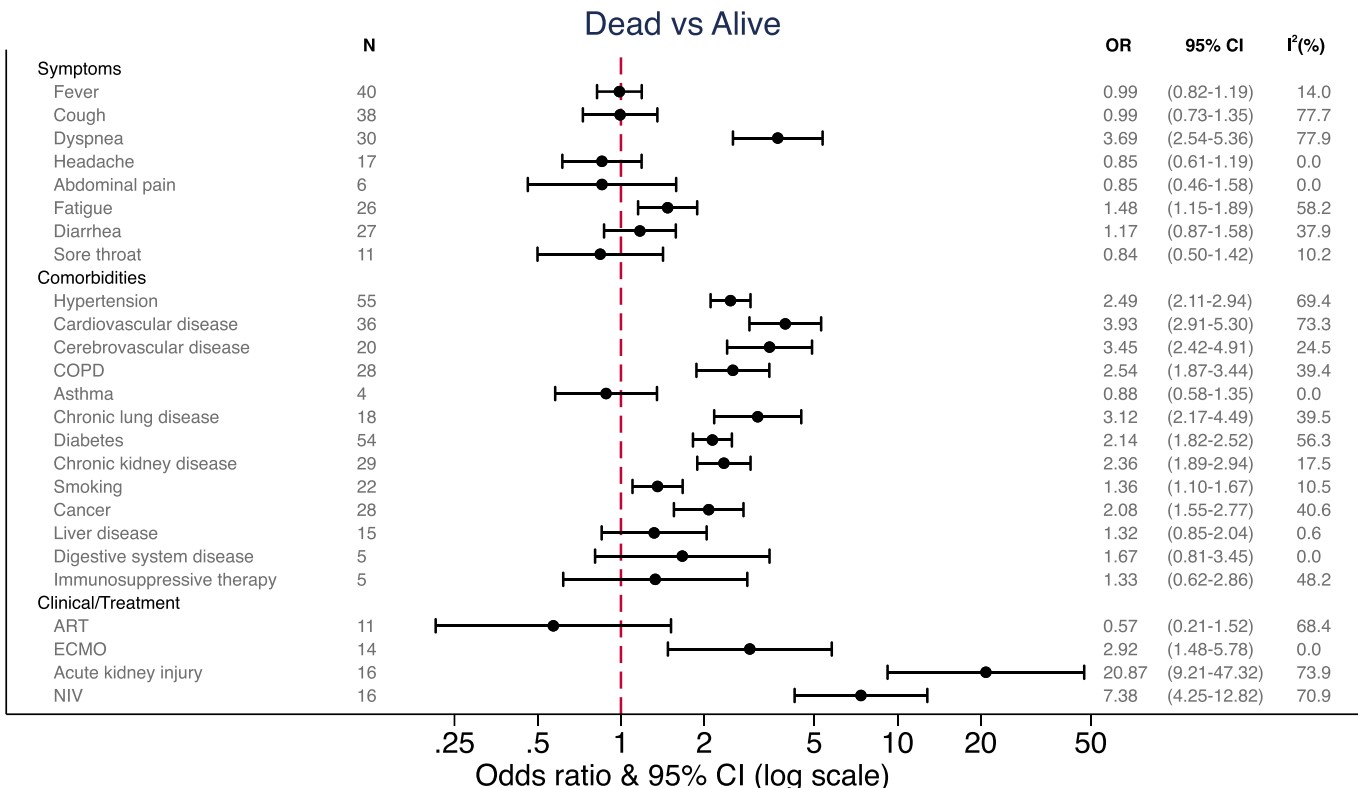

**Fig 4. Pooled odds ratios among mortality vs. survived groups.** OR = odds ratio, CI = confidence interval, COPD = chronic obstructive pulmonary disease, ART = anti-retroviral treatment, ECMO = extracorporeal membrane oxygenation, NIV = non-invasive ventilation.

As a clinical complication, acute kidney injury showed the highest overall odds ratio for mortality (OR 20.87, CI 9.21 to 47.32), followed by requiring non-invasive ventilation (NIV) (OR 7.38, CI 4.25 to 12.82).

Fig 5 shows pooled median estimates along with their normal laboratory ranges for selected number of indicators among patients who died, patients who survived, ICU-admitted patients, and non-ICU admitted patients. Pooled difference of medians estimates for all indicators are available in the supplementary files (S5 for mortality in S1 File, S6 for ICU admission in S1 File, S7 for intubation and hospitalization in S8 in S1 File). After removing large outliers in a sensitivity analyses for CRP and D-dimer, results did not change substantially (results available in the supplementary S9 in S1 File). Funnel plots showed no substantial asymmetry suggesting small-study effects except for data assessing acute kidney injury (supplementary S10 for mortality and S11 for ICU admission in S1 File).

## Discussion

In this comprehensive systematic review and meta-analysis, we corroborate known markers of severe disease for COVID-19 and shed light on further indicators, whose significance was indeterminate to date.

With respect to co-morbidities, we identified cardiovascular disease, which includes chronic heart disease and coronary artery disease (OR 3.93), chronic lung disease (OR 3.12) and COPD (OR 2.54) as strong risk factors of mortality among COVID-19 patients but not for ICU admission. Only cerebrovascular disease was strongly associated with an increased risk of both ICU admission and death (almost six- and three-fold higher ratio for ICU admission and

**Table 2. Summary of the meta-analysis results for continuous indicators comparing those who died and those who survived.**

| Indicator | N. Studies | Pooled DoM [95% CI] | I² |
|---|---|---|---|
| **Demographics** | | | |
| Age (years) | 52 | 13.15 [11.37, 14.94] | 86.74 |
| **Clinical Values** | | | |
| SpO2—without O2 (%) | 15 | -6.33 [-8.14, -4.52] | 81.77 |
| Respiratory Rate (per min) | 15 | 3.41 [2.26, 4.55] | 62.32 |
| **Laboratory Values** | | | |
| Hemoglobin (g/L) | 18 | -2.66 [-5.12, -0.2] | 43.36 |
| Leukocyte (10⁹/L) | 37 | 2.79 [2.23, 3.35] | 70.35 |
| Lymphocyte (10⁹/L) | 38 | -0.34 [-0.39, -0.29] | 70.03 |
| Neutrophil (10⁹/L) | 25 | 3.26 [2.56, 3.95] | 82.2 |
| Platelets (10⁹/L) | 30 | -31.94 [-41.11, -22.77] | 58.13 |
| APTT (sec) | 16 | 0.59 [-0.51, 1.69] | 61.88 |
| D-Dimer (mg/L)* | 30 | 1.29 [0.90, 1.69] | 81.53 |
| Fibrinogen (g/L) | 7 | 0.01 [-0.12, 0.15] | 0.00 |
| INR | 7 | 0.06 [0.01, 0.12] | 63.31 |
| Prothrombin (sec) | 25 | 0.91 [0.67, 1.14] | 54.65 |
| ALAT (U/L) | 34 | 4.43 [2.41, 6.46] | 26.64 |
| Albumin (g/L) | 21 | -4.64 [-5.83, -3.45] | 85.16 |
| ASAT (U/L) | 27 | 13.35 [10.54, 16.15] | 42.83 |
| LDH (U/L) | 23 | 189.49 [155, 223.98] | 75.03 |
| BUN (mmol/L) | 17 | 2.77 [2.07, 3.46] | 66.77 |
| Creatinine (μmol/L) | 29 | 15.3 [10.3, 20.29] | 61.63 |
| CRP (mg/L)* | 34 | 69.1 [50.43, 87.77] | 95.99 |
| IL-6 (pg/mL) | 11 | 31.19 [11.96, 50.41] | 99.75 |
| PCT (ng/mL) | 18 | 0.16 [0.1, 0.22] | 68.09 |
| BNP (pg/mL) | 7 | 405.26 [116.51, 694.02] | 95.81 |
| CK (U/L) | 18 | 64.09 [29.04, 99.13] | 81.47 |
| CK-MB (U/L) | 9 | 3.66 [1.19, 6.14] | 67.12 |
| cTnI (pg/mL)* | 13 | 21.88 [9.78, 33.99] | 75.17 |

*Indicates that the DL approach was used to estimate between-study heterogeneity. SpO2 = Oxygen saturation; APTT = activated partial thrombin time; INR = Internationalized normalized ratio; ALAT = Alanine transaminase; ASAT = Aspartate transaminase; LDH = Lactate dehydrogenase; BUN = Blood urea nitrogen; CRP = C-reactive protein; IL-6 = Interleukin-6; BNP = brain natriuretic peptide; PCT = Procalcitonin CK = creatine kinase; CK-MB = creatine kinase–myocardial band; TnI = cardiac Troponin I.

death, respectively). Overall, the finding that cerebrovascular disease is associated with poor outcomes is in line with the more recent data highlighting the importance of delirium and an overall depressed mental state in severe COVID-19 [111–113]. Our findings found chronic kidney disease, diabetes [18] and COPD/chronic lung disease to be risk factors, however, associations are less strong than those for cardiovascular or cerebrovascular disease [111]. Evidence from previous studies regarding the risk associated with hypertension has been inconclusive. Our work identifies hypertension as a clear risk factor for ICU admission (OR 1.62, CI 1.24 to 2.12) and death (OR 2.49, CI 2.11 to 2.94) [7, 19, 20]. Similarly, while prior data were inconclusive with respect to the influence of smoking for severe COVID-19 [3, 22–25], our meta-analysis shows the increased risk of mortality among smokers (OR 1.36, CI 1.10 to 1.67). However,

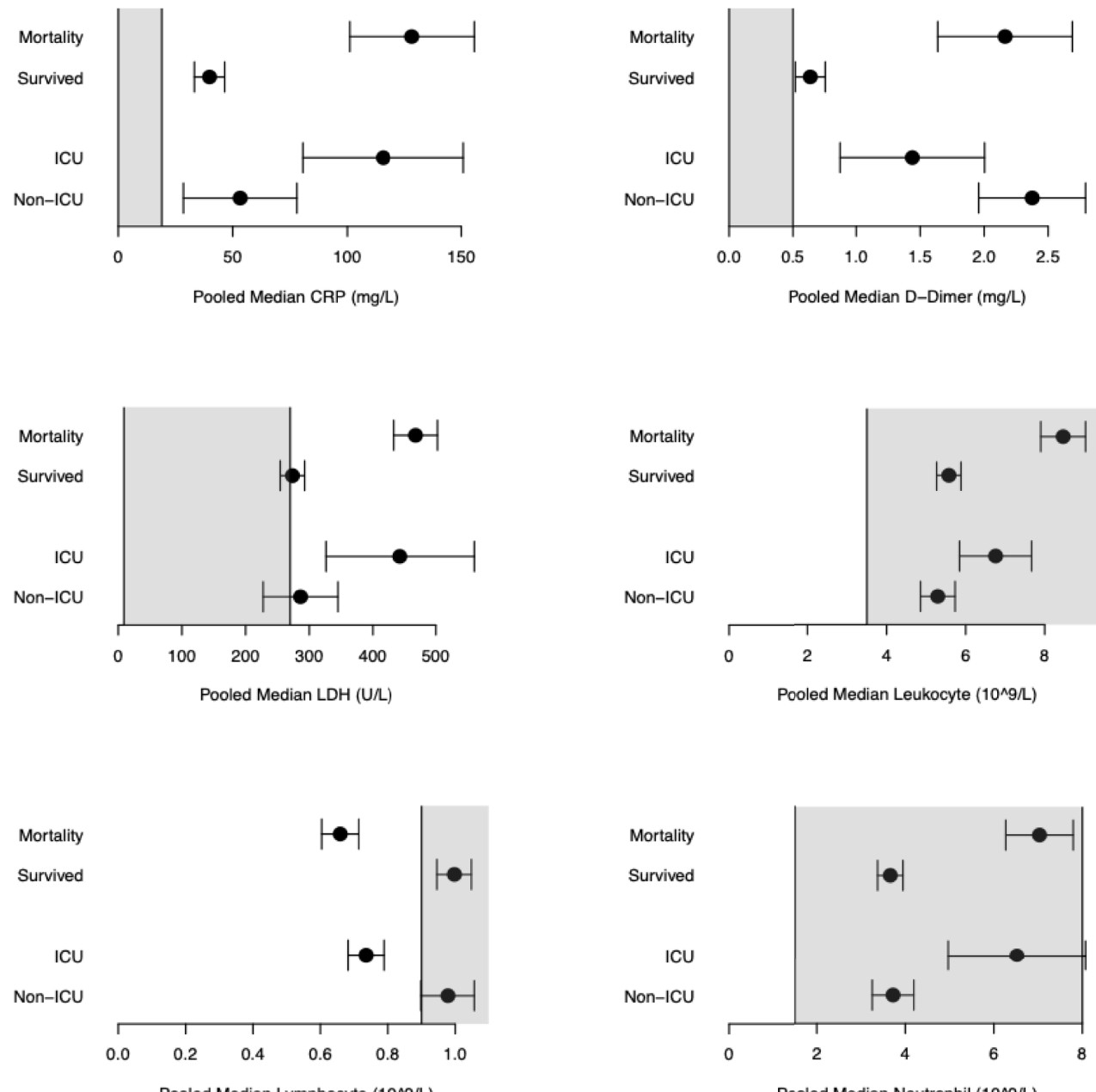

**Fig 5. Pooled median estimates of selected indicators along with their normal laboratory ranges among patients who died, patients who survived, ICU-admitted patients, and non-ICU admitted patients.** ICU = Intensive care unit, CRP = C-reactive protein, LDH = Lactate dehydrogenase.

our data did not allow for meta-regression to assess whether this effect was independent of the risk associated with chronic lung disease. In line with some recent studies on asthma, we could not find an increased risk for mortality [114] in our meta-analysis (OR 0.88, CI 0.58 to 1.35).

CRP was the only laboratory marker that was associated with a higher risk of ICU admission (DoM 56.41 mg/L) and death (DoM 69.1 mg/L), while D-dimer elevation was only

significantly associated with death (DoM 1.29 mg/L), but not with ICU admission. Although the median elevation of cTnI was clinically relevant both in those who were admitted to the ICU (DoM 19.27 pg/mL) and those who died (DoM 21.88 pg/mL), only in those who died was the difference statistically significant.

We were able to identify clinically relevant lymphopenia as a marker. Lymphopenia was a marker that was used early on for triage purposes to predict disease severity [115] and our findings on increased odds for mortality corroborate the systematic review results on this topic published by Huang and Pranata [116].

For categorical variables, OR was used to measure the association between the outcome (mortality, ICU admission, hospitalization, intubation) and the risk factor/biomarker of interest. We decided to use the OR instead of the relative risk (RR) given the nature of the research question and the data that were available to us. For example, calculating the risk of ICU among people with fever compared to the risk of ICU among people without a fever is not as informative as computing the odds ratio of ICU admission between fever and non-fever patients. We acknowledge, however, when interpreting the results that the ORs are more extreme (further from the null) than the RRs whenever there is a non-null association.

In line with previous reviews, we acknowledge risk factors such as age, dyspnea, smoking, diabetes, hypertension and cardiovascular disease to be associated with increased odds for mortality.

On the one hand we could not find an increased odds for asthma, whereas 'respiratory diseases' [3] had an increased OR in other reviews. On the other hand, we find COPD to be a strong risk factor for mortality. This can be explained by different data extractions and pooling of similar diseases. Nevertheless, this suggests that lung diseases are associated with an increased risk of a severe course.

## Strengths and limitations of this study

Our study provides a comprehensive review of the data from both pre-print and peer-reviewed sources with a broad geographic distribution and assesses the different categories of risk factors from symptoms, co-morbidities and laboratory values to clinical complications. Correlating the indicators to the two clinical outcomes death and ICU admission, has both strengths and limitations. While ICU admission is a clinical decision, it is, especially early on in a new disease, sometimes a measure of precaution. This might weaken the association of indicators with clinical outcomes. At the same time, when capacity of ICU beds is exhausted, triage decisions might have been made based on age and co-morbidities to not admit to the ICU, thus strengthening an association of an indicator beyond what would be expected under routine conditions. We also assessed the association with hospitalization and intubation (see supplement), but here confounding factors seemed to be even more pronounced, and data are further limited. Confounding factors that lead to this conclusion are for example different health care systems across the globe with or without the possibility of self-care in less severe cases instead of hospital admission or the change in the approach regarding early intubation from the first wave towards a more conservative approach with novel methods such as 'awake proning'.

In addition, with improving care and novel therapies certain associations might be less pronounced. We did not observe improved survival with antiviral therapy in the studies included, suggesting that this effect might not yet have occurred in the time frame of studies included here. We have not assessed radiological findings as these would likely correlate with clinical signs and symptoms, as well as changes in laboratory parameters. In contrast to other systematic reviews, we did not focus on the course of the disease (e.g. critical, severe), but rather on the outcome [3]. Furthermore risk factors were assessed for each outcome individually [17].

Additional limitations primarily relate to data quality of the included studies. Our quality assessment of studies clearly indicated that substantial bias was present across studies. Primarily the selection bias as suggested for example by the high case fatality rate (e.g. Zhou et al. [11], 28.3%, Chen et al. [117] 11.1%, and Huang et al. [12] 14.6%) is likely to have impacted our results and prospective data collection to confirm findings of these studies is important [118]. However, an analysis of quality was performed at the study level. Conceivably a high-quality study will contribute more high quality data on the risk-factor outcome association, but this cannot be ascertained. In addition, 13 studies were still in preprint at the time of extraction. Furthermore, we found a large number of studies (n = 21, list available in supplementary S3 in S1 File) to include laboratory values that were clinically out of range, which suggests that despite peer-review in some of them, the rush of publication in this pandemic impacted the quality of reporting [119].

## Conclusion

Our data on mortality and ICU admission corroborate most of the proposed indicators of clinical outcomes, clarifies the strength of association and highlights additional indicators. In addition, this systematic review highlights the limitations of the studies published and calls for better quality in prospective collections.

## Supporting information

**S1 PRISMA checklist.**
(DOCX)

**S1 File. Supplementary file with supporting information.** This contains supporting information for all outcomes with summary forest plots for 'intubation' and 'hospitalization', risk of bias assessment and funnel plots.
(DOCX)

## Acknowledgments

We thank Genevieve Gore for her help with the search terms. We thank Sinan Onogur, Laura Aguilera Saiz, and Bünyamin Pekdemir for their help setting up the data extraction tool.

## Author Contributions

**Conceptualization:** Stephan Katzenschlager, Claudius Gottschalk, Jürgen Grafeneder, Alexander Seitel, Andrea Benedetti, Sean McGrath, Claudia M. Denkinger.

**Data curation:** Stephan Katzenschlager, Claudius Gottschalk, Jürgen Grafeneder, Stephani Schmitz, Sara Kraker, Marlene Ganslmeier, Amelie Muth, Alexander Seitel, Lena Maier-Hein, Sean McGrath, Claudia M. Denkinger.

**Formal analysis:** Stephan Katzenschlager, Alexandra J. Zimmer, Stephani Schmitz, Sara Kraker, Marlene Ganslmeier, Amelie Muth, Sean McGrath, Claudia M. Denkinger.

**Investigation:** Claudia M. Denkinger.

**Methodology:** Stephan Katzenschlager, Alexandra J. Zimmer, Claudius Gottschalk, Jürgen Grafeneder, Alexander Seitel, Lena Maier-Hein, Andrea Benedetti, Sean McGrath, Claudia M. Denkinger.

**Project administration:** Stephan Katzenschlager, Claudia M. Denkinger.

**Resources:** Claudia M. Denkinger.

**Software:** Alexandra J. Zimmer, Lena Maier-Hein, Sean McGrath, Claudia M. Denkinger.

**Supervision:** Stephan Katzenschlager, Claudia M. Denkinger.

**Validation:** Stephan Katzenschlager, Alexandra J. Zimmer, Claudia M. Denkinger.

**Visualization:** Stephan Katzenschlager, Alexandra J. Zimmer, Claudia M. Denkinger.

**Writing – original draft:** Stephan Katzenschlager, Alexandra J. Zimmer, Sean McGrath, Claudia M. Denkinger.

**Writing – review & editing:** Stephan Katzenschlager, Alexandra J. Zimmer, Claudius Gottschalk, Jürgen Grafeneder, Alexander Seitel, Lena Maier-Hein, Andrea Benedetti, Jan Larmann, Markus A. Weigand, Sean McGrath, Claudia M. Denkinger.

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
