## [Decision Letter · Decision Letter 0]

25 May 2021

PONE-D-20-41054

Can we predict the severe course of COVID-19; a systematic review and meta-analysis of indicators of clinical outcome?

PLOS ONE

Dear Dr. Denkinger,

Thank you for submitting your manuscript to PLOS ONE. After careful consideration, we feel that it has merit but does not fully meet PLOS ONE’s publication criteria as it currently stands. Therefore, we invite you to submit a revised version of the manuscript that addresses the points raised during the review process.

The topic is quite intersting in clinical practice and the paper well written. We suggest to provide an accurate revision of language. For istance to use relative risks instead of odds ration for categorical variabiles. 

We look forward to receiving your revised manuscript.

Kind regards,

Chiara Lazzeri

Academic Editor

PLOS ONE

Journal Requirements:

3) Please include the date(s) on which your research team performed the seaaches.

4)  Thank you for stating the following in the Acknowledgments/Funding Section of your manuscript:

[SM acknowledges support from the National Science Foundation Graduate Research Fellowship Program

under Grant No. DGE1745303, National Library Of Medicine of the National Institutes of Health under

Award Number T32LM012411, and Fonds de recherche du Québec-Nature et technologies B1X research

scholarship.

CMD acknowledges the support of the Heidelberg University Hospital and the German Center of Infectious

Disease, where Heidelberg is a core site as part of the tuberculosis focus group.

The work was supported by Heidelberg University Hospital internal funds]

 [The study was supported by internal funds of the Heidelberg University Hospital. The

funders play no role in the study design, data collection and analysis, decision to

publish, or preparation of the manuscript.]

5) We note that you have stated that you will provide repository information for your data at acceptance. Should your manuscript be accepted for publication, we will hold it until you provide the relevant accession numbers or DOIs necessary to access your data. If you wish to make changes to your Data Availability statement, please describe these changes in your cover letter and we will update your Data Availability statement to reflect the information you provide.

6) Please include a copy of Table 3 which you refer to in your text on page 10.

Reviewers' comments:

Reviewer's Responses to Questions

**Comments to the Author**

1. Is the manuscript technically sound, and do the data support the conclusions?

Reviewer #1: Yes

2. Has the statistical analysis been performed appropriately and rigorously? 

Reviewer #1: Yes

3. Have the authors made all data underlying the findings in their manuscript fully available?

Reviewer #1: Yes

4. Is the manuscript presented in an intelligible fashion and written in standard English?

Reviewer #1: Yes

5. Review Comments to the Author

Reviewer #1: Thanks for the opportunity of reviewing this manuscript,

The aim of the paper is to systematically evaluate the literature reporting on demographic, clinical and laboratory risk factors for disease severity.

I think this paper is a considerable effort to create a synthesis of the Covid-19 risk factors for disease severity. However, before being accepted for publication I have three overall main comments that I think the authors should consider.

- The risk of bias evaluation was conducted and reported at the study level for a study that is focus on the evidence regarding individual risk factor-outcome level across different studies. There lies a limitation in the sense that it is hard to know what is the confidence/uncertainty surrounding each risk factor-outcome association.

- The categorical variables were presented in terms of odds ratios instead of relative risks. Odds ratios are a valid measure of epidemiological association, however there might be a risk of overestimating estimates relative to estimates calculated in terms of relative risks. I think it is worth commenting on the likelihood of this overestimation based on the authors knowledge of the literature and if there was a particular rationale supporting the use of odds ratios.

- In light of the above presented ideas, I would suggest changing any comments affirming that the study “confirms” most of the proposed indicators. My suggestion comes from the idea that in English language the term confirm seem to be more associated with establishing the truth or correctness of something, similar to beweisen (in German). Perhaps the authors refer confirm as bestätigen (in German) to convey validate, affirm, reaffirm, certify. It is a different claim to suggest findings support previous estimates, than to present them as an established scientific truth, especially without conducting an evidence quality analysis at the outcome level.

Additional suggestions by section:

Abstract:

-Methods: I would suggest reporting the PROSPERO registration in the methods section of the abstract.

-Results: I would suggest to include P values in abstract

- Discussion: By decision analytical tools are you referring to development of clinical scores for predicting outcomes?

Introduction:

Perhaps add that it was October 31 2020 or October 31 of the same year. It seems redundant but it reads weirdly for me with just the month and the day. As a reader, I got distracted thinking about it.

Acknowledge the existence of previous reviews and systematic reviews discussing risk factors for covid-19 severity and highlight what is the added value of the present review:

e.g.

Risk factors of critical & mortal COVID-19 cases: A systematic literature review and meta-analysis.

Zheng Z, Peng F, Xu B, Zhao J, Liu H, Peng J, Li Q, Jiang C, Zhou Y, Liu S, Ye C, Zhang P, Xing Y, Guo H, Tang W.J Infect. 2020 Aug;81(2):e16-e25. doi: 10.1016/j.jinf.2020.04.021. Epub 2020 Apr 23.

A brief-review of the risk factors for covid-19 severity

JE Rod, O Oviedo-Trespalacios, J Cortes-Ramirez

Rev Saude Publica. 54 (60)

Methods:

-Eligibility criteria: Please provide a brief comment on the rationale for focusing on the presented outcomes for severity e.g.: why other markers of severity as O2 saturation or the development of acute respiratory distress syndrome where not included.

-Study and screening: “Article title and abstracts were screened for eligibility in (instead of and?) English and German”.

-Assessment of study quality: Study quality might be a more important step in aiming at ranking the studies in a way that could perhaps exclude some studies and include only studies with high quality rankings. Here it seems that the intention was to provide a comment on the risk of bias of each study. I mention this given that the individual evaluation of the risk of bias for different outcomes (hospital admission, intubation. etc) across different papers does not necessarily share the same risk of bias as the whole paper. Eg. some papers report bivariate analysis for one set of outcomes but multivariate analysis for other outcomes.

I understand that conducting a risk of bias for each of the outcomes is extremely labor some. However, it must be acknowledged that the risk of bias assessment was performed at the study and not the outcome level and therefor the risk of bias assessment might work as an index of a risk of bias at the outcome level, but deeper analyses might yield a different result.

If the study is focus on conducting meta-analysis at multiple outcomes levels,

-Statistical analysis: Perhaps, but not necessarily, it would be clearer to separate study outcomes in “primary” and “secondary” based on data availability and said that secondary outcomes comparisons can be found in the supplement. This could also be reflected on the results section.

If hypertension is not considered cardiovascular disease, could you please specify what diseases where included under the umbrella term “cardiovascular disease”.

Discussion:

I think there should be a brief paragraph contrasting current findings with those of previous reviews and why the authors think there might be similarities or contrast.

Assessing clinical relevance? Does statistically significant result mean automatically clinically relevant?

Limitations:

Consider mentioning:

- Missing radiological findings as risk factors

- When it is mentioned that: “We also assessed the association with hospital and intubation…, but here confounding factors seem to be even more pronounced”. I got confused because confounding factors where not mentioned previously in the text for the other outcomes. This might imply that the author knows what they mean about the confounding factors of the other outcomes, but this is not mentioned in the text.

I think something like this should be mentioned:

- The aim is to analyze individual outcomes. However, quality analysis was performed at the study and not the outcome level. This imply that the conclusion of the evidence quality analysis might be valid, but not necessarily so. More granular analysis of the evidence outcome level might yield different results. It is possible that this is not the case, but I think it is important to mentioned for methodological reasons.

Conclusions:

I think it is a bit overconfident to suggest that the review “confirms” the previous and new indicators as risk factors for covid-19 severity. Please do not take this comment as an attempt to undermine the academic value of the contribution. Both systematic review and meta-analysis are considered the highest quality of evidence in health-related research. However, the review did not evaluate evidence quality at the outcome level. This method is specifically designed to evaluate evidence quality (and not only risk of bias) at the outcome level and use this information to develop clinical guidelines.

I would suggest that the word “confirm” should be replaced by support.

6. PLOS authors have the option to publish the peer review history of their article (what does this mean?). If published, this will include your full peer review and any attached files.

Reviewer #1: No

---

## [Author Response · Author response to Decision Letter 0]

3 Jul 2021

Thank you for the opportunity to respond to the reviewers' and editors' comments. We have attached a rebuttal letter with a point-by-point response. We hope you find this satisfactory.

---

## [Editor Report · Decision Letter 1]

12 Jul 2021

Can we predict the severe course of COVID-19; a systematic review and meta-analysis of indicators of clinical outcome?

PONE-D-20-41054R1

Dear Dr. Denkinger,

We’re pleased to inform you that your manuscript has been judged scientifically suitable for publication and will be formally accepted for publication once it meets all outstanding technical requirements.

Kind regards,

Chiara Lazzeri

Academic Editor

PLOS ONE
---

## [Editor Report · Acceptance letter]

22 Jul 2021

PONE-D-20-41054R1 

Can we predict the severe course of COVID-19  – a systematic review and meta-analysis of indicators of clinical outcome? 

Dear Dr. Denkinger:

I'm pleased to inform you that your manuscript has been deemed suitable for publication in PLOS ONE. Congratulations! Your manuscript is now with our production department. 

Kind regards, 

on behalf of

Dr. Chiara Lazzeri 

Academic Editor

PLOS ONE